# The P2X7 Receptor as a Mechanistic Biomarker for Epilepsy

**DOI:** 10.3390/ijms24065410

**Published:** 2023-03-12

**Authors:** Tobias Engel

**Affiliations:** 1Department of Physiology and Medical Physics, RCSI, University of Medicine and Health Sciences, D02 YN77 Dublin, Ireland; tengel@rcsi.ie; Tel.: +353-14025199; 2FutureNeuro, Science Foundation Ireland Research Centre for Chronic and Rare Neurological Diseases, RCSI, University of Medicine and Health Sciences, D02 YN77 Dublin, Ireland

**Keywords:** epilepsy, seizures, diagnosis, treatment, purinergic signalling, P2X7 receptor

## Abstract

Epilepsy, characterized by recurrent spontaneous seizures, is a heterogeneous group of brain diseases affecting over 70 million people worldwide. Major challenges in the management of epilepsy include its diagnosis and treatment. To date, video electroencephalogram (EEG) monitoring is the gold-standard diagnostic method, with no molecular biomarker in routine clinical use. Moreover, treatment based on anti-seizure medications (ASMs) remains ineffective in 30% of patients, and, even if seizure-suppressive, lacks disease-modifying potential. Current epilepsy research is, therefore, mainly focussed on the identification of new drugs with a different mechanism of action effective in patients not responding to current ASMs. The vast heterogeneity of epilepsy syndromes, including differences in underlying pathology, comorbidities and disease progression, represents, however, a particular challenge in drug discovery. Optimal treatment most likely requires the identification of new drug targets combined with diagnostic methods to identify patients in need of a specific treatment. Purinergic signalling via extracellularly released ATP is increasingly recognized to contribute to brain hyperexcitability and, consequently, drugs targeting this signalling system have been proposed as a new therapeutic strategy for epilepsy. Among the purinergic ATP receptors, the P2X7 receptor (P2X7R) has attracted particular attention as a novel target for epilepsy treatment, with P2X7Rs contributing to unresponsiveness to ASMs and drugs targeting the P2X7R modulating acute seizure severity and suppressing seizures during epilepsy. In addition, P2X7R expression has been reported to be altered in the brain and circulation in experimental models of epilepsy and patients, making it both a potential therapeutic and diagnostic target. The present review provides an update on the newest findings regarding P2X7R-based treatments for epilepsy and discusses the potential of P2X7R as a mechanistic biomarker.

## 1. Introduction and Shortcomings in Epilepsy Treatment

Epilepsy is one of the most common chronic brain diseases, affecting up to 70 million people worldwide. It has an incidence rate of approximately 1–2%, which is higher in low-middle-income countries compared to high-middle-income countries [1,2,3]. While affecting people of all ages, epilepsy is most common among the young and elderly, with a slightly higher prevalence in men compared to women. Major challenges of epilepsy include its diagnosis and treatment, social disadvantages (e.g., unemployment, stigma) and the increased risks of premature mortality (up to 3-fold) and co-morbidities (up to 8-fold), such as depression and anxiety [1,2,4,5]. This imposes a particular high burden on patients and makes societal costs of epilepsy one of the highest for any neurological disease [6].

Epilepsy can be acquired after an injury to the brain (e.g., traumatic brain injury (TBI), stroke, infection, an episode of prolonged seizures (e.g., status epilepticus)) or can be caused by genetic abnormalities (e.g., polymorphisms, copy number variations or de novo mutations) [7,8]. Epileptogenesis, triggered following a precipitating injury to the brain, is the process of turning a normal healthy brain into a brain experiencing epileptic seizures. Pathological changes occurring during epileptogenesis and beyond the occurrence of the first epileptic seizure include ongoing neurodegeneration, aberrant synaptic plasticity and neurogenesis, blood–brain barrier (BBB) disruption and inflammation [7,9]. Among these, mechanisms driving neuroinflammatory processes have attracted particular attention as potential drug targets [10]. Temporal lobe epilepsy (TLE), which can be acquired after a brain injury, is the most common form of epilepsy in adults and involves structures within the limbic system, including the amygdala and hippocampus [11]. The most common pathological finding in the brain of TLE patients is hippocampal sclerosis, characterized by selective neuronal loss and reactive gliosis [12]. Of note, TLE remains the most common studied epilepsy type in the setting of purinergic signalling, in particular involving the P2X7R receptor, as discussed within the following sections.

Frontline treatment for epilepsy is based on anti-seizure medications (ASMs), which are mainly focused on targeting synaptic transmission (glutamate, gamma-aminobutyric acid (GABA) and ion channels (Na^+^/Ca^2+^ channels)) [13]. However, despite the availability of over 25 ASMs in clinical use, drug refractoriness among patients remains at approximately 30%. Moreover, even if effective in suppressing seizures, ASMs have no proven effect in modifying disease progression and can cause serious side effects (e.g., drowsiness, agitation, headaches) [2,14]. There is, therefore, a pressing need to identify new druggable targets with a different mechanism of action with treatments effective in drug-refractory patients, with disease-modifying potential and causing less adverse side effects.

In addition to difficulties in treatment, the diagnosis of seizures and epilepsy is complex and remains a clinical challenge. The diagnosis of epilepsy requires at least two unprovoked seizures occurring more than 24 h apart, including cases with one seizure and a high likelihood of having additional seizures (more than 60%) or the diagnosis of an epilepsy syndrome. Epilepsy can be classified into three levels: seizure type, epilepsy and epilepsy syndrome, with a strong emphasis on etiology and comorbidities [15,16,17]. To date, the gold standard for epilepsy diagnosis is with long-term electroencephalogram (EEG) video recordings. This is, however, very costly, has low throughput and is not always available as primary healthcare [17]. This not only causes difficulties in identifying people with epilepsy but also differentiating epilepsy syndromes from similar conditions, such as convulsive syncope or psychogenic non-epileptic attacks [18,19]. This can further lead to wrong or unnecessary treatments with the additional risk of drug-induced adverse side effects. Moreover, there are no markers to identify patients at risk of developing epilepsy after, for example, an injury to the brain (e.g., TBI). There is, therefore, a strong need to develop easy-to-use and affordable diagnostic methods (e.g., biomarkers), which would not only support the diagnosis of epilepsy but also assess disease progression. Epilepsy biomarkers would also help in the monitoring of therapeutic effects, therapeutic trial designs and decision making when prescribing seizure-suppressive or anti-epileptogenic drugs and could be used to identify patients who would most likely benefit from a specific treatment (personalized medicine).

Purinergic signalling via adenosine triphosphate (ATP) is increasingly recognized as a new therapeutic target for epilepsy. This includes ATP receptors, such as the P2X7 receptor (P2X7R) [20], as discussed in detail within this review, and other components of the ATP signalling cascade, including ATP release mechanisms (e.g., Pannexin-1 channels) [21] or ATP-metabolizing enzymes (e.g., tissue-nonspecific alkaline phosphatase (TNAP)) [22]. Notably, data also suggest a diagnostic potential in the purinergic signalling system for seizures and epilepsy [23], making this signalling system particularly suited for the management of epilepsy and its treatment.

## 2. ATP Signalling via the P2X7R

Over the past decade, ATPergic signalling via purinergic P2 receptors has established itself as a major player in epilepsy pathology, with increasing data showing its effectiveness as a novel therapeutic target [24,25]. Aside from its role in metabolism, ATP acting as a signalling molecule was proposed in the seminal work by Geoffrey Burnstock in 1972 [26]. To date, signalling mediated via purinergic P2 receptors has been described to be involved in mostly all cellular processes in the brain, including modulating neurotransmission, cell death, proliferation and differentiation, neuron–glia communication and neuroinflammation [27,28,29,30].

The purinergic signalling system encompasses the synthesis and release of purine nucleosides and nucleotides (e.g., adenosine, ATP), purinergic receptors, as well as the machinery to eliminate purine molecules from the extracellular space [31]. Usually present at low extracellular concentrations (nanomolar range), extracellular ATP concentrations can rise dramatically during pathological conditions into the millimolar range [32], where ATP acts as an endogenous ‘danger’ signal, known as damage-associated molecular patterns (DAMPs) [33]. ATP can be released via exocytotic mechanisms, such as via the Cl^−^-dependent vesicular nucleotide transporter (VNUT) [34,35], or via non-exocytotic mechanisms, including voltage-dependent anion channels [36], ATP-binding cassette transporters [37,38], purinergic P2 receptors (e.g., P2X7R [39]) or via channels, such as connexins and pannexins [40,41] or, due to its huge concentration gradient between the intra- and extracellular space, passively through damaged cell membranes [42,43].

Evidence suggesting increased extracellular ATP concentrations following seizures and during epilepsy stems from studies using acute recordings from brain slices and measurements carried out in in vivo models of epilepsy (e.g., [44,45,46,47,48,49,50,51]). Here, a study by Dona et al. reported increased extracellular ATP concentrations following epileptic seizures using a rat model of intraperitoneal (i.p.) pilocarpine. Interestingly, while other purines, such as adenosine 5′-diphosphate (ADP), adenosine monophosphate (AMP) and adenosine, were also increased extracellularly post-status epilepticus, this was not the case for ATP [49]. Likewise, no increase in extracellular ATP levels has been observed following pilocarpine-induced status epilepticus in rats [48]. Progress has also been made in the identification of ATP release mechanisms during seizures, with a particular focus on the Pannexin-1 channel [21]. This includes evidence from a study where rat hippocampal slices were treated with a glutamate agonist ((S)-3,5-Dihydroxyphenylglycine) [47]. In another study, which was carried out in resected tissue from patients, Dossie et al. [21] showed that extracellular ATP concentrations increased by 80% during high K^+^-induced ictal discharges, which were suppressed by blocking Pannexin-1. Of note, the same authors showed potent anticonvulsive effects when blocking Pannexin-1 during kainic acid (KA)-induced seizures in mice [21].

ATP-sensitive purinergic receptors include the cationic, trimeric P2X membrane receptor (P2XR) family and the metabotropic P2Y membrane receptors (P2YRs) [52,53]. The P2XR family comprises seven subunits (P2X1-7), with each P2XR subunit consisting of two membrane-spanning domains (TM1 and TM2), with an intracellular N- and C-terminus. P2XRs form a trimeric ligand-gated ion channel that can be either homomeric or heteromeric according to the recruitment of identical or diverse receptor subtypes. P2XRs are all activated by ATP, thereby allowing for the passage of positively charged ions, such as Na^+^, Ca^2+^ and K^+^ [54]. The metabotropic P2YR family is composed of eight subunits (P2Y_1,2,4,6,8,11,12,14_) that have the typical seven-transmembrane segment of G-protein-coupled receptors, which are activated by ATP (P2Y_2_, P2Y_11_), ADP (P2Y_1_, P2Y_12_ and P2Y_13_), uridine 5′-triphosphate (UTP)/uridine 5′-diphosphate (UDP) (P2Y_2_, P2Y_4_ and P2Y_6_) and UDP-glucose (P2Y_14_). For a more detailed description of P2YRs, please refer to [52,55] or other reviews written on this topic.

While data generated over the past decade have demonstrated a role for both P2X and P2YRs during seizures and epilepsy [24,25,56,57,58], most studies have focused, by far, on P2XRs, particularly on P2X7R. Initially termed ATP4− or P2Z receptor and characterized on cells of the immune system, such as mast cells, lymphocytes or macrophages [59,60,61], P2X7R is now recognized to be expressed and functional throughout the body, including the brain. There are several reasons making the P2X7R a particularly attractive therapeutic target for brain diseases such as epilepsy. Among the P2XRs, P2X7R has a relatively low affinity for ATP (EC50 ≥ 100 μM, activation threshold: 0.3–0.5 mM), suggesting P2X7R activation occurs mainly under conditions of high ATP release, restricted to the pathological focus. As a result, P2X7R-based treatments are hoped to lead to less adverse side effects when compared to current treatments (e.g., ASMs). Most importantly, however, P2X7R is a key driver of inflammation [62,63,64]. P2X7Rs have been shown to contribute to the activation and proliferation of microglia [65] and are key regulators of the nucleotide-binding oligomerization domain-, LRR- and pyrin domain-containing protein 3 (NLRP3) inflammasome, inducing the release of inflammatory signalling molecules, such as the pro-inflammatory cytokine interleukin-1β (IL-1β) [66,67,68,69]. P2X7Rs are expressed throughout the central nervous system (CNS), including brain areas previously implicated in epilepsy development, such as the frontal cortex, hippocampus and amygdala, where they have been reported to be functional on all cell types. However, while its expression on microglia and oligodendrocytes is well-established, whether P2X7Rs are expressed on neurons and astrocytes remains a matter of debate [70,71]. P2X7Rs participate in numerous intracellular signalling pathways in addition to inflammation activated during epilepsy, including the modulation of neurotransmitter release (e.g., glutamate), cell death, BBB opening and synaptic plasticity [71,72,73,74,75]. It is, however, mainly their effects on inflammation and microglia activation [65] that have generated strong interest, including from academia and industry. This has resulted in the generation of new transgenic models, important for a better understanding of P2X7R-mediated signalling during normal physiology and disease (e.g., enhanced green fluorescent protein (EGFP)-P2X7R reporter mice [76]), and the development of several highly specific, BBB-permeable and brain-stable P2X7R antagonists [77,78], critical to advance P2X7R-based treatments towards a clinical application for brain diseases.

## 3. The Role of P2X7Rs during Seizures and Epilepsy

Over the past few decades, a substantial body of evidence has been accumulated demonstrating a role for P2X7R during seizure generation and epilepsy with data stemming from both experimental models of epilepsy and patients [24,57]. More recent data also show the diagnostic potential of P2X7R signalling [23]. Since several reviews on the therapeutic potential of targeting P2X7R during seizures and epilepsy have been published recently (e.g., [24,57,58]), the present review will mainly focus on the diagnostic potential of P2X7R signalling for epilepsy. This will, however, be preceded by a brief description on what is known regarding the cell-type-specific expression changes in P2X7Rs during epilepsy and the anticonvulsant and anti-epileptic potential of drugs targeting P2X7Rs.

### 3.1. P2X7R Expression following Seizures and during Epilepsy

P2X7R expression has been analysed following acute seizures (i.e., status epilepticus) and during epilepsy in brain tissue of several experimental rodent models and human brain tissue. While broad consensus exists regarding its overall expression changes (i.e., up-regulation), less agreement exists regarding its cell-type-specific expression. P2X7R expression has consistently been found to be increased at the transcriptional and protein level in the hippocampus and cortex following pilocarpine (i.p.) and KA (i.p., intra-amygdala, intra-hippocampal)-induced status epilepticus in rats and mice, shortly following status epilepticus and during epilepsy [79,80,81,82,83,84,85,86,87]. However, while studies using i.p. KA-treated mice and pilocarpine-treated rats have suggested P2X7R to be up-regulated mainly on microglia shortly following status epilepticus [80,88], other studies using different seizure models, including i.p. pilocarpine in rats and intra-amygdala KA in mice, suggested a neuronal up-regulation in the hippocampus (e.g., mossy fibres, granule cells of the dentate gyrus) and cortex [81,83,84,85]. This also includes a study where status epilepticus was triggered via intra-amygdala KA in 10-day-old rat pups [89]. P2X7R expression was also found to be increased on neuronal progenitor cells following status epilepticus induced via i.p. pilocarpine in mice [90]. Neuronal expression of P2X7R has, however, been questioned using a P2X7R reporter mouse, where P2X7R is fused to EGFP [76]. Here, P2X7R-EGFP was expressed on microglia and oligodendrocytes post-intra-amygdala KA; however, no co-localization was observed for P2X7R-EGFP with neurons [87]. No P2X7R expression was found on astrocytes [87]. Of note, using the same P2X7R-EGFP reporter mouse, Morgan et al. found widespread changes in P2X7R expression following intra-amygdala KA-induced status epilepticus throughout the brain (acute and epilepsy), suggesting that the P2X7R acts as a common pathological factor in the brain, possibly contributing to the widespread inflammation and damage seen in TLE [87]. Evidence of P2X7R expression changes also occurring in humans stems from studies analysing resected brain tissue from drug-refractory TLE patients after epilepsy surgery. This confirmed increased P2X7R expression in the hippocampus [85,91,92] and cortex [84]. The cell-type-specific expression pattern in human epilepsy has, however, not been analysed to date.

With regard to the mechanisms which regulate P2X7R expression in the brain during seizures, a study carried out by us using the intra-amygdala KA mouse model suggested P2X7R transcription being, at least partly, under the control of the specificity protein 1 (Sp1) [86]. P2X7R expression seems to be, however, also controlled at the post-transcriptional level, involving the targeting of *P2rx7* mRNA by microRNA-22, which is also under the control of Sp1 [86]. Here, we showed that elevated Ca^2+^ concentrations, due to status epilepticus, blocked the binding of Sp1 to the promoter of microRNA-22, thereby disinhibiting the suppression of *P2rx7* mRNA translation into protein, leading, in turn, to increased P2X7R expression [86].

In summary, while it is well established that P2X7R expression increases on microglia post-status epilepticus and during epilepsy, where it possibly contributes to pro-inflammatory signalling in the brain, whether P2X7R expression also changes in astrocytes and neurons remains to be determined.

### 3.2. The Role of P2X7R Signalling during Seizures and Epilepsy

Over the past decade, numerous studies have investigated the anticonvulsant, antiepileptogenic and antiepileptic potential of drugs targeting P2X7R (Table 1). As mentioned before, in the majority of cases, the models of choice were models which mimic TLE and where epilepsy development is triggered via chemically induced prolonged damaging seizures (i.e., status epilepticus). This includes models where status epilepticus is triggered via a systemic injection of pilocarpine or KA or models where KA is injected directly into the brain (e.g., intra-amygdala, intra-hippocampal). Other, less frequent models used were models of acute, non-damaging seizures. This includes the pentylenetetrazol (PTZ) model and models where seizures are elicited via electrical stimulation (e.g., 6 Hz model and electroshock seizure threshold). A more detailed description of experimental models used for the study of P2X7R signalling during epilepsy can be found here [93,94,95,96,97].

P2X7R antagonism has been shown to reduce seizure severity in the intra-amygdala and intrahippocampal KA mouse model and the coriaria lactone-induced status epilepticus mouse model [83,84,101,105]. On the contrary, P2X7R antagonism has been shown to promote seizures in the i.p. pilocarpine model of status epilepticus [90,111]. In contrast to seizures, however, P2X7R antagonism has been shown to protect against seizure-induced cell death and reduce neuroinflammation in both KA-induced and pilocarpine-induced status epilepticus models [83,90,112]. Notably, P2X7R antagonism also reduced seizure severity in a mouse model of hypoxia-induced neonatal seizures [98,106] and a model of early life seizures in rat pups [89], suggesting that the observed anticonvulsive effects are not age-dependent. No, or only weak, effects of P2X7R antagonism were observed on non-damaging seizures. This included the PTZ seizure threshold, maximal electroshock seizure threshold and 6 Hz psychomotor seizure threshold test [104,108]. Likewise, no effects of P2X7R antagonism on seizures were observed in WAG/Rij rats, an inbred strain with genetic absence epilepsy [102].

In contrast to acute seizures, results from models of epileptogenesis and epilepsy are more consistent with P2X7R antagonism, typically reducing seizure severity or suppressing seizures. P2X7R antagonism decreased the mean kindling score using the PTZ kindling model in rats [108,110], and treatment with P2X7R-targeting siRNA delayed the emergence of the first seizure and reduced the frequency and severity of seizures in the i.p. pilocarpine rat model [103]. In addition, P2X7R antagonisms applied after hypoxia-induced seizures in mouse pups reduced long-lasting brain hyperexcitability [98], further suggesting anti-epileptogenic potential. In the same line, P2X7R antagonism during epilepsy reduced seizure severity without altering the frequency of seizures in a model where epilepsy was induced via i.p. KA in rats [107]. Using the intra-amygdala KA mouse model, we showed that P2X7R antagonism reduced the total number of seizures during treatment in epileptic mice. These effects persisted beyond drug withdrawal, implying disease-modifying potential [85]. Of note, previous research has shown that mice, where epilepsy was induced via intra-amygdala KA, are partially resistant to ASMs [114], suggesting P2X7R antagonists as a possible treatment option for drug-refractory epilepsy. Further evidence supporting a role for P2X7R during epilepsy development stems from a study where mice were treated with inhibitors against microRNA-22, previously shown to target *P2rx7* mRNA [109]. These mice presented increased hippocampal P2X7R expression and developed a more severe epileptic phenotype. This was accompanied by increased cytokine release and astrogliosis [109]. Finally, a recent study analysed the epileptic phenotype in mice transgenic for the lysosomal enzyme palmitoyl protein thioesterase 1 (PPT1). These mice develop epilepsy after 7 months of age, including increased microgliosis [99]. When treated with P2X7R antagonists, PPI transgenic mice experienced fewer seizures, further supporting the antiepileptogenic potential of P2X7R antagonism. Taken together, while both anti- and proconvulsive effects of P2X7R antagonism on acute seizures have been reported, broader consensus exists regarding its effects on epilepsy development and established epilepsy where the main finding was P2X7R antagonism-mediated seizure suppression.

Unresponsiveness to ASMs remains one of the most pressing challenges in the treatment of status epilepticus and epilepsy, with increasing evidence suggesting neuroinflammatory pathways contributing to drug refractoriness [115]. In line with P2X7R contributing to unresponsiveness to ASMs, we have recently shown that mice overexpressing P2X7Rs were less responsive to several anticonvulsants (lorazepam, midazolam, phenytoin and carbamazepine) during status epilepticus [100]. Suggesting these effects to be mediated via P2X7R driving inflammation, in the same study, we showed that (i) P2X7R expression was increased in microglia during drug-refractory status epilepticus, (ii) microglia in P2X7R-overexpressing mice presented a pro-inflammatory phenotype during status epilepticus and (iii) that the anti-inflammatory drug minocycline restored normal responsiveness to anticonvulsants in P2X7R-overexpressing mice [100]. In addition, and further suggesting these effects to be mediated via inflammation, pre-treatment with the pro-inflammatory agent lipopolysaccharide (LPS) increased not only P2X7R expression in the brain but also decreased the responsiveness of mice to anticonvulsants. LPS-induced drug refractoriness was overcome via a genetic deletion of P2X7R and via treatment with P2X7R antagonists [100], suggesting P2X7R-based treatments as adjunctive treatment for drug-refractory status epilepticus and, possibly, drug-refractory epilepsy. Notably, further suggesting the potential of P2X7R antagonists as adjunctive treatment for pharmaco-resistant status epilepticus, previous studies have shown that P2X7R antagonists, when given in combination with lorazepam at a timepoint during intra-amygdala KA-induced status epilepticus when sensitivity to lorazepam was reduced, efficiently stopped seizures [83]. Likewise, P2X7R antagonists, when given in combination with carbamazepine, increased the seizure threshold in the maximal electroshock test [108].

The molecular pathways of how P2X7R signalling impacts on seizures and epilepsy remain to be established. While P2X7R driving inflammatory processes is the most likely explanation [116], it is important to keep in mind that P2X7R signalling has been involved in numerous pathways in the brain, with several of these having a known role during epileptogenesis (e.g., maintenance of the BBB, synaptic plasticity, neurogenesis, control of neurotransmitter release, et cetera) [71,72,73,74,75,117]. Evidence supporting P2X7R contributing to brain hyperexcitability via driving inflammation include that (i) P2X7Rs are highly expressed on microglia, which increases following status epilepticus and increases during epilepsy [87,100], (ii) blocking of P2X7Rs during status epilepticus reduced hippocampal IL-1β levels [83] (iii) and the fact that while microglia from P2X7R-overexpressing mice showed a pro-inflammatory phenotype during status epilepticus [100], microglia from P2X7R KO mice presented an anti-inflammatory phenotype following hypoxia-induced seizures in neonatal mouse pups [98]. In addition, treatment of mice with an anti-inflammatory agent (i.e., minocycline) abolishes P2X7R-mediated effects during seizures [98,100]. If pro-convulsant effects of P2X7Rs are mediated via driving inflammation, this may also partly explain differences in P2X7R-based treatment responses observed between different models and disease stages. In this scenario, for P2X7R antagonists to reduce brain hyperexcitability, this would require underlying inflammation, which may differ between models (e.g., status epilepticus vs. acute non-damaging seizures; acute pathology vs. ongoing pathology during epilepsy). Other possible explanations include substrate availability. P2X7Rs require elevated extracellular ATP concentrations [118], potentially only available under more severe conditions (e.g., status epilepticus) or ongoing pathology (i.e., epilepsy). Cell-type-specific responses according to the model used may further contribute to the observed differences. While highly expressed on microglia following seizures and during epilepsy, P2X7Rs have also been reported to be increased on oligodendrocytes and neurons [81,85,87]. Whether expression changes in these cell types contribute to the epileptic phenotype remains, however, to be shown. Therefore, much more research is needed to identify the exact pathways of how P2X7Rs contribute to epilepsy via, for example, the use of cell-type-specific P2X7R KO mice in microglia or neurons.

### 3.3. The P2X7R as Mechanistic Biomarker for Seizures and Epilepsy

The capacity to identify patients who are in need of and can benefit most from treatments (personalized medicine) is an important ability to improve the efficiency of pharmacological approaches. This is even more important in the treatment of diseases as heterogeneous as epilepsies, where patients are likely to show different responses according to the underlying aetiology and pathology. Moreover, mechanistic biomarkers, with a known role during disease pathogenesis, are much more informative and more accurately reflect the disease state compared to descriptive biomarkers, which result as a side product of the disease. As outlined before, P2X7R expression has been shown to be altered in brain tissue during epilepsy (mice and humans) [57]. In addition, P2X7R expression is not restricted to the brain and P2X7Rs are highly expressed on different circulating immune cells, where they contribute to the activation of the peripheral immune system and the release of inflammatory molecules, such as the known P2X7R downstream pro-inflammatory cytokine IL-1β, into the circulation [119,120]. Of note, several P2X7R-dependent inflammation markers have been shown to be dysregulated in the blood of epilepsy patients (e.g., IL-1β [121,122], IL-18 [123]), potentially serving as useful surrogate makers of P2X7R activation/inhibition. Critically, diagnostic tools capable of detecting P2X7R protein and P2X7R downstream molecules are already available, including P2X7R-based positron emission tomography (PET) radioligands [124] and assays able to detect P2X7Rs in fluids such as blood (ELISA) [120] (Table 2).

#### 3.3.1. P2X7R-PET Imaging as Novel Diagnostic Tool for Epilepsy

PET imaging is a well-established technique and routinely used for the diagnosis of brain diseases, including epilepsy [130]. To date, PET imaging in epilepsy is primarily used to image glucose metabolism via the radiotracers 2-[18F]fluoro-2-deoxy-D-glucose ([18F]fluorodeoxyglucose or [18F]FDG [131]. However, over the past few years, radiotracers detecting other molecules have been developed with radiotracers recognizing inflammatory targets, such as the Translocator protein (TSPO), a suggested marker of activated glia [132,133], being of particular interest.

Several P2X7R radiotracers have now been successfully developed and tested in experimental models of epilepsy and resected brain tissue from patients, including the P2X7R-specific radioligand [^18^F]JNJ-64413739 [134]. Using this radioligand, a recent study analysing human cortical tissue sections resected from a total of 48 drug-resistant epilepsy patients demonstrated specific binding of [^18^F]JNJ-64413739 in different brain structures, including grey and white matter [125]. When correlating P2X7R-PET radioligand uptake in tissue sections to clinical and demographic data from patients, the authors found, however, no significant correlation of P2X7R radiotracer uptake with age, sex and duration of epilepsy. It is, however, noteworthy to mention that P2X7R radioligand uptake was not correlated to other clinical data, such as underlying pathology (e.g., presence or absence of hippocampal sclerosis) or disease severity and seizure frequency. Of note, P2X7R-radiotracer uptake showed an almost inverse correlation to the widely used surrogate marker of glial density, TSPO, measured via [^123^I]CLINDE, suggesting that either both proteins are expressed in different cell types and/or that their regulation differs according to disease stage. In a second recent study, Fu et al. investigated P2X7R radiotracer dynamics via PET/computerized tomography (CT) longitudinal imaging using the P2X7R-PET ligand ^18^F-FTTM in a rat model, where status epilepticus was induced via an intracranial injection of KA into the CA1 subfield of the hippocampus [127]. P2X7R-PET imaging was carried out at three different disease stages: (i) acute (1–2 days post-KA injection), (ii) latent period (7–10 days post-KA) and (iii) established epilepsy (3 months post-KA). Increased P2X7R radiotracer uptake was evident immediately post-status epilepticus and peaked during the latent period returning to baseline control levels during epilepsy. This included several brain structures, such as the hippocampus, amygdala and temporal cortex. The authors further concluded that this increase was mostly related to microglia, suggesting that changes in P2X7R-PET ligand uptake may serve as a readout of altered brain inflammation during epilepsy [127].

Further exploring the diagnostic potential of P2X7R-PET imaging for epilepsy, we recently published a study using the P2X7R radiotracer [^18^F]JNJ-64413739 [126]. Here, P2X7R radiotracer uptake was measured 48 h post-intra-amygdala KA-induced status epilepticus in mice, a timepoint that represents the seizure-free latent period in this model [135]. While P2X7R radiotracer uptake was not increased when comparing mice subjected to status epilepticus with control, we found a strong correlation between P2X7R radiotracer uptake and the severity of status epilepticus, with mice undergoing a more severe status epilepticus showing higher P2X7R radioligand brain levels when compared to mice with a milder status epilepticus and control. Of note, P2X7R radiotracer uptake was evident throughout the brain, including both ipsi- and contralateral brain regions (e.g., hippocampus, cortex, amygdala and thalamus), similar to findings from Fu et al. [127] and to our results using P2X7R-EGFP reporter mice [87]. While the diffuse P2X7R expression throughout the brain most likely means that P2X7R-based PET imaging is not a good tool to identify the epileptic focus, the fact that P2X7R radioligand uptake correlated with the severity of status epilepticus suggests that P2X7R-PET may have potential as a predictive biomarker supporting the stratification of patients according to disease progression and underlying pathology. Noteworthily, increased P2X7R expression has been shown to reduce the responsiveness to current ASMs [100]. Measuring P2X7R radioligand uptake may, therefore, not only help to identify patients possibly benefitting from P2X7R-based treatments but also to identify patients at risk of drug refractoriness. Of note, we observed a similar correlation between P2X7R radiotracer uptake and seizure severity during status epilepticus in several organs. This was most evident in the liver, heart and lungs. Epilepsy is characterized by increased systemic inflammation [136] and comorbidities that involve peripheral organs such as the heart [137]. Future studies should be designed to establish whether P2X7Rs contribute to epilepsy-induced organ damage and whether this can be prevented/mitigated via drugs targeting this receptor. Finally, in the same study, we found increased P2X7R radiotracer uptake in resected tissue sections from epilepsy patients when compared to control, further confirming the diagnostic potential of P2X7R-based PET. Whether P2X7R-PET has the potential to stratify patients beyond epilepsy diagnosis (i.e., underlying pathology, epilepsy syndrome, disease severity) will, however, require the analysis of much larger patient cohorts.

In summary, while still at an early stage and more detailed studies in animal models and in larger epilepsy patient cohorts are required, P2X7R-based PET imaging may represent a promising tool to support the diagnosis of patients at risk of/with epilepsy and to identify patients who would benefit from P2X7R-based treatments.

#### 3.3.2. P2X7R Signalling Components as Diagnostic Tools for Seizures and Epilepsy

P2X7R expression is not restricted to brain tissue, and P2X7Rs can be found throughout the body, including the peripheral immune system (e.g., macrophages, T cells) [119,138,139]. Suggesting its potential as a diagnostic marker, a recent study by Giuliani et al. showed that P2X7Rs can be shed into circulation, which seems to be increased under inflammatory conditions. The same study identified monocytes as the most likely source of P2X7Rs [120]. Subsequent studies have shown increased P2X7R in blood cells of patients with diabetes [140] and myasthenia gravis [141]. P2X7R protein levels were also increased in the plasma of patients with sepsis and COVID-19 [142,143].

To test whether P2X7R expression changes in the blood of patients with epilepsy, in a recent study, we analysed plasma from TLE patients (n = 30) via ELISA and compared these to plasma samples from patients suffering from psychogenic non-epileptic seizures (PNES) (n = 11), patients who underwent a recent episode of status epilepticus (n = 6) and healthy controls (n = 34) [128]. PNES patients show similar behaviour changes resembling an epileptic seizure, but without the characteristic alterations on the EEG associated with epileptic seizures, and are a particularly valuable control group, as most are on a similar treatment regimen as patients with epilepsy [144]. Plasma samples were analysed during baseline (i.e., following an at least 24 h seizure-free period) and 1 h after a recent seizure. This revealed increased P2X7R plasma concentrations in patients with TLE and in patients with status epilepticus when compared to control and patients with PNES. No further increases to baseline levels were observed 1 h following seizures, suggesting that increased P2X7R plasma levels are due to the underlying pathology rather than being a direct result of seizures. While differences in P2X7R plasma concentrations between TLE patients and control were only moderate (sensitivity (60%) and specificity (74%)), ROC analysis between TLE patients and patients with PNES showed a high sensitivity (90%) and good specificity (63%). Of note, in contrast to previous studies [145], no significant difference between groups was found when analysing the inflammation marker C-reactive protein (CPR). In order to identify what blood cells overexpress P2X7R following seizures, EGFP-P2X7R reporter mice were subjected to intra-amygdala KA-induced status epilepticus and EGFP-positive blood cells quantified via FACS. This showed increased EGFP signal in blood post-status epilepticus and identified white blood cells as the most likely cell type to overexpress P2X7Rs, similar to what had been shown in humans [120]. To identify P2X7R downstream signalling in blood and, thereby, possible markers of P2X7R over-activation, blood samples were analysed via cytokine arrays from wild-type and P2X7R knock-out mice subjected to intra-amygdala KA-induced status epilepticus. This identified the cytokine keratinocyte chemoattractant/human-growth-regulated oncogene (KC/GRO) as a potential P2X7R-dependent plasma biomarker following status epilepticus and during epilepsy [128].

Genetic alterations (e.g., de novo mutations, genetic polymorphism) are increasingly recognized to contribute to epilepsy and even to unresponsiveness to ASMs [146]. The human *P2rx7* gene has been described to be highly polymorphic, with several single-nucleotide polymorphisms (SNPs) known to change the receptor function into either loss- or gain-of-function variants [62,63]. While mainly associated to other brain diseases, such as major depression, mood disorders and sleep disorders [58,147,148], a study published by Emsley et al. in 2014 suggests the gain-of-function missense SNP rs208294 in *P2rx7* to be involved in susceptibility to childhood-onset febrile seizures [129]. Whether P2X7R SNPs are a risk for epilepsy development, unresponsiveness to ASMs or disease severity has, to my knowledge, not been published yet.

Finally, while not directly related to P2X7R signalling, but nevertheless demonstrating the diagnostic potential of the purinergic signalling system for epilepsy, blood purines measured via summated electrochemical detection of adenosine and adenosine breakdown products inosine, hypoxanthine and xanthine were elevated after status epilepticus in mice and in TLE patients (baseline levels) [149]. Blood purine levels in mice correlated with seizure severity during status epilepticus and seizure-induced neurodegeneration, suggesting blood purines as a possible tool supporting patient stratification according to underlying pathology. Interestingly, we also showed blood purines to be increased in mouse pups after hypoxia-induced seizures and infants with neonatal encephalopathy and seizures [150]. Whether P2X7R contributes to changes in blood purines during seizures and epilepsy remains to be determined. ATP can, however, also be released via P2X7Rs [39] and, once released, is rapidly broken down into different breakdown products including adenosine [151].

## 4. Conclusions

We now have a substantial body of evidence demonstrating not only the therapeutic potential of targeting the P2X7R during epilepsy but also the proof of concept of its diagnostic capabilities, making this receptor, without doubt, an increasingly promising therapeutic target (Figure 1). While there are still important issues, which will have to be addressed in future studies, including testing P2X7R antagonists in human tissue and the use of larger patient cohorts, with P2X7R antagonists and P2X7R-PET radiotracers already at the clinical trial stage [124,152], it can be hoped that P2X7R-based treatments for epilepsy will reach the clinic within the foreseeable future.

## Figures and Tables

**Figure 1 ijms-24-05410-f001:**
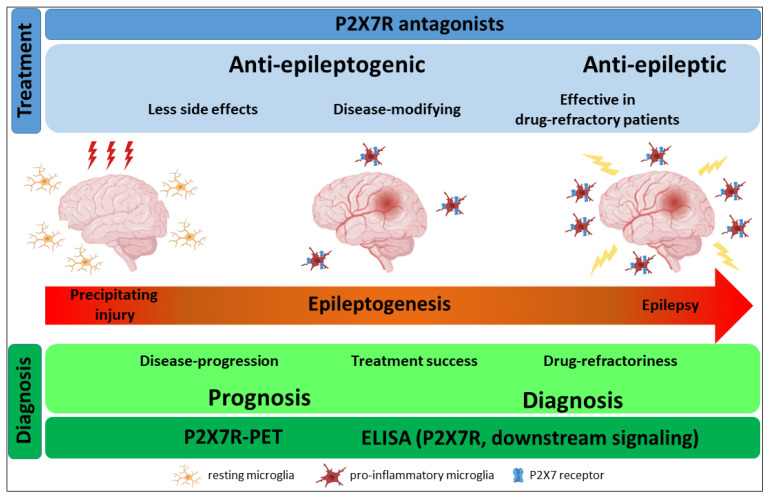
P2X7R as a therapeutic and diagnostic target for epilepsy. Following a precipitating injury to the brain, P2X7R expression increases on microglia where it promotes pro-inflammatory signalling, thereby contributing to increased brain hyperexcitability and epilepsy development. P2X7R antagonism mitigates the development of epilepsy and suppresses seizures during epilepsy, possibly leading to disease modification. Diagnostic tools based on P2X7R include P2X7R radiotracers and assays to measure P2X7R plasma concentrations and P2X7R downstream signalling molecules.

**Table 1 ijms-24-05410-t001:** Selected studies investigating the therapeutic potential of targeting P2X7R during seizures and epilepsy.

Experimental Approach to Induce Seizures/Epilepsy	Approaches to Manipulate/Visualize P2X7R	Changes in P2X7R Expression	Effects on Seizures, Epilepsy and Pathology	Reference
*Neonatal seizures and epileptogenesis* hypoxia-induced seizures (5% O_2_, 15 min) in male and female P7 mouse pups	P2X7R antagonist JNJ-47965567 (30 mg/kg, i.p.) P2X7R KO mice EGFP-P2X7R overexpressing mice	P2X7R mainly localized to microglia and oligodendrocytes post-hypoxia.	P2X7R KO decreased and P2X7R overexpression increased seizure severity during hypoxia; P2X7R overexpression increased unresponsiveness to ASMs; P2X7R KO promoted anti-inflammatory phenotype in microglia; P2X7R antagonism post-hypoxia reduced long-lasting brain hyperexcitability.	[98]
*Epilepsy* Lysosomal enzyme palmitoyl protein thioesterase 1 (PPT1) knock-in mice (c.451C > T/c.451C > T) (sex not specified)	P2X7R antagonist A-438079 (30 mg/kg, twice a day for two days, i.p.)	Not analysed.	Reduction in microglia numbers in hippocampus. Reduction in duration and number of seizures in PPT1 KO mice via P2X7R antagonism.	[99]
*Status epilepticus* intra-amygdala KA (0.3 µg (C57/Bl6) and 0.2 µg (FVB)) in male and female mice	P2X7R antagonists AFC-5128 (50 mg/kg, i.p.) and ITH15004 (1.75 nmol, i.c.v.) P2X7 KO mice EGFP-P2X7R overexpressing mice	Increased P2X7R expression in microglia during SE.	Increased P2X7R expression (EGFP-P2X7R and post-LPS treatment) reduces responsiveness to ASMs during SE most likely via P2X7R promoting neuroinflammation; P2X7R KO/antagonism restores responses to ASMs in drug-refractory models of SE.	[100]
*Status epilepticus* intrahippocampal KA (4 µg) in male rats	P2X7R antagonist BBG (2 nM) 30 min prior to KA via intracerebral infusion	Not analysed.	Decreased seizure severity, astrogliosis, mossy fibre sprouting and neuronal death, and improved spatial memory via P2X7R antagonism.	[101]
*Absence seizures* WAG/Rij male rats (inbred strain of *rats* with genetic absence epilepsy)	P2X7R agonist B_Z_ATP (50 and 100 μg, i.c.v) and P2X7R antagonist A-438079 (20 μg, i.c.v.)	Not analysed.	No effects of P2X7R agonists or antagonists on spike-wave discharges (SWDs).	[102]
*Status epilepticus and epileptogenesis * i.p. pilocarpine (370 mg/kg) in male and female rats; i.p. pilocarpine in male and female mice (300 mg/kg)	P2X7R antagonists AZ10606120 (1 µL/min flow rate, 3 µg in total, i.c.v.) and BBG (50 mg/kg, i.p.) *P2rx7*-GFP reporter mice	Increased P2X7R currents at neural progenitor cells (NPCs) in the subgranular zone of the dentate gyrus (mice).	P2X7R antagonisms (AZ10606120) prevented neurodegeneration in CA3 but increased seizure number and seizure severity during epilepsy (rats).	[90]
*Status epilepticus and epileptogenesis * i.p. pilocarpine (370 mg/kg) in male rats	P2X7R-targeting siRNA (i.c.v.)	Not analysed.	P2X7R antagonisms mediated neuroprotection in hippocampus, reduced edema, reduced mortality following SE, delayed seizure onset and seizure numbers during epilepsy.	[103]
*Focal, generalized and generalized tonic-clonic* timed i.v. PTZ infusion test, MES-T and 6 Hz electroshock-induced seizures in mice	P2X7R antagonist BBG (50–200 mg/kg, i.p.) for i.v. PTZ and MES-T test and 25–100 mg/kg, once daily for seven consecutive days for 6 Hz test	Not analysed.	Reduced seizures during 6 Hz test (focal seizure) via BBG. No significant anticonvulsive effects of BBG in i.v. PTZ and MES-T test (generalized and generalized tonic-clonic seizures).	[104]
*Status epilepticus* intra-muscular coriaria lactone (40 mg/kg) in male rats	P2X7R antagonists BBG (1 μg, 5 μg and 10 μg, i.c.v.) and A-740003 (10 μM, i.c.v.), and P2X7R agonist BzATP (5 mM, i.c.v.)	Increased hippocampal P2X7R levels 1–2 days post-SE which gradually decreased to baseline by 2 weeks post-SE.	P2X7R antagonism reduced neuronal damage, inflammation (astrogliosis and microgliosis), seizures and improved cognitive function.	[105]
*Neonatal seizures* hypoxia-induced seizures (5% O_2_, 15 min) in male and female P7 mouse pups	P2X7R antagonists A-438079 (0.5, 5, 15, 25 and 50 mg kg^−1^ i.p.) and JNJ-47965567 (10 and 30 mg/kg, i.p.)	Increased P2X7R expression in hippocampus and cortex.	P2X7R antagonism reduced caspase-1 processing, microglia numbers and seizure severity.	[106]
*Epilepsy* multiple low-dose i.p. KA (5 mg/kg KA, repeated every hour until SE was established) in male rats	P2X7R antagonist JNJ-47965567 (0.6 g/kg/2 mL, s.c.) via osmotic mini-pump for 7 days	Not analysed.	Decreased seizure severity, but no changes in the total number of seizures. P2X7R antagonist did not alter microglia activation or astrogliosis.	[107]
*Epilepsy* intra-amygdala KA (0.3 µg) in male mice	P2X7R antagonist JNJ-47965567 (20 mg/kg, i.p.) *P2rx7*-GFP reporter mice	Increased P2X7R expression in hippocampus during epilepsy localized mainly to microglia and neurons. P2X7R increased in TLE patient brain (hippocampus).	P2X7R antagonism-mediated reduction in frequency of spontaneous seizures which was evident beyond drug-withdrawal. Decreased inflammation (astrogliosis and microgliosis).	[85]
*Acute seizures and epileptogensis* MES-T (sinusoidal pulses 4–14 mA, 50 Hz, 0.2 s duration) and PTZ-T (87 mg/kg s.c.) in male mice; i.p. PTZ kindling (35 mg/kg, i.p. once every 48 h and 3 times a week) in male rats	P2X7R antagonists JNJ-47965567 (15 and 30 mg/kg s.c.), AFC-5128 (25 and 50 mg/kg s.c.), BBG (50 mg/kg i.p.) and TIIAS (tanshinone IIA-SO3Na) (30 mg/kg i.p.)	Colocalisation of P2X7R immunofluorescence with microglia-like cells and synaptophysin in the PTZ kindling model.	No effects of P2X7R antagonism on acute seizures (MES-T and PTZ-T test). In the PTZ kindling model, AFC-5128- and JNJ-47965567 reduced Iba1 and GFAP immunoreactivity in the hippocampus. Moreover, AFC-5128 and JNJ-47965567 showed a significant and long-lasting delay in kindling development. P2X7R antagonism potentiates effects of ASM carbamazepine.	[108]
*Epileptogensis* intra-amygdala KA (0.3 µg) in male mice	microRNA-22 targeting antagomir (2 μL infusion of 0.5 nmol, i.c.v.)	Increased P2X7R expression in the ipsilateral hippocampus and reduced P2X7R expression in the contralateral hippocampus post-SE.	Increased astrogliosis and microgliosis in antagomir-22 treated epileptic mice. Antagomir-22 treated mice develop more severe epileptic phenotype.	[109]
*Epileptogensis* i.p. PTZ kindling in rats (30 mg/kg) every other day for 27 days (14 injections)	P2X7R antagonist BBG (15 and 30 mg/kg, i.p.)	Not analysed.	P2X7R antagonism via BBG decreased mean kindling score and improved motor performance and cognitive deficits.	[110]
*Early life seizures* intra-amygdala KA (2 μg in 0.2 μL) in 10-day-old rat pups	P2X7R antagonist A-438079 (0.5, 5, 15, and 50 mg/kg, i.p.)	Increased P2X7R expression in the hippocampus and co-localization to mossy fibres.	P2X7R antagonism reduced seizure severity and seizure-induced neurodegeneration.	[89]
*Status epilepticus* intra-amygdala KA (0.3 µg) in male mice	P2X7R antagonist A438079 (0.75 nmol, i.c.v.) *P2rx7*-GFP reporter mice	Increased P2X7R expression in cortex of mice post-SE and during epilepsy localized to neurons and microglia. Increased P2X7R expression in cortex of TLE patients.	P2X7R antagonism reduced seizure severity during SE.	[84]
*Status epilepticus* intra-amygdala KA (3 µg) in male mice	P2X7R agonist B_Z_ATP (10.5 nmol, i.c.v.), P2X7R antagonists A438079 (1.75 nmol, i.c.v.) and BBG (1 pmol, i.c.v.) and P2X7R antibody (0.7 mg/mL, i.c.v.) P2X7R KO mice *P2rx7*-GFP reporter mice	Increased P2X7R expression in hippocampus. P2X7R mainly localized to neurons with some microglial expression.	Reduced IL-1β levels and Iba-1-positive microglia numbers in hippocampus due to P2X7R antagonism. P2X7R agonist-increased seizure severity, while P2X7R antagonists/P2X7R KO reduced seizure severity during SE. Reduced neurodegeneration via P2X7R antagonism. P2X7R antagonism potentiates effects of anticonvulsant lorazepam.	[83]
*Status epilepticus* i.p. pilocarpine (150, 175, 200, 225, or 250 mg/kg), i.p. picrotoxin (5 mg/kg) and i.p. KA (25 mg/kg) in male mice	P2X7R agonist BzATP (5 mM) and P2X7R antagonists OxATP (5 mM), A-438079 (10 μM) and A740003 (10 μM) via osmotic mini-pumps (3 days) P2X7R KO mice	Not analysed.	P2X7R deletion and blockade increased pilocarpine-induced seizure susceptibility via non-glutamatergic and non-GABAergic transmission. No effects of P2X7R KO on seizures in i.p. KA and i.p. picrotoxin model.	[111]
*Status epilepticus* i.p. pilocarpine (380 mg/kg) in male rats	P2X7R agonist B_Z_ATP (5 mM) and P2X7R antagonists OxATP (5 nM) and BBG (5 nM) via osmotic mini-pump	Not analysed.	BzATP increased TNF-α immunoreactivity in dentate granule cells, which was decreased via OxATP. P2X7R antagonism reduced astroglial death in the molecular layer of the dentate gyrus and the frontoparietal cortex, however, promoted clasmatodendrosis in CA1.	[112]
*Status epilepticus* i.p. pilocarpine (380 mg/kg) in male rats	infusion of P2X7R agonist BzATP (5 mM, 43 μg, i.c.v.) and P2X7R antagonist OxATP (5 mM, 30 μg, i.c.v.) via osmotic mini-pump	Not analysed.	P2X7R antagonism reduced infiltration of neutrophils into the frontoparietal cortex.	[113]
*Status epilepticus* i.p. KA (18–22 mg/kg) in mice (sex not specified)	P2X7R antagonist BBG (3 µM) in brain slices	Increased *P2rx7* mRNA in hippocampus.	Increased microglia membrane currents via P2X7R 48 h after SE (this also includes P2Y_6_R and P2Y_12_R).	[82]

Abbreviations: BBG, brilliant blue G; BzATP, 2’(3’)-O-(4-Benzoylbenzoyl)adenosine-5’-triphosphate; i.p., intraperitoneal; i.c.v., intracerebroventricular; i.v., intravenous; KA, Kainic acid; KO, Knock out; MES-T, Maximal electroshock seizure threshold test; OxATP, adenosine 5’-triphosphate-2’,3’-dialdehyde; PTZ-T, Pentylenetetrazol seizure threshold test; SE, Status epilepticus.

**Table 2 ijms-24-05410-t002:** Selected studies investigating the diagnostic potential of P2X7R for seizures and epilepsy.

Seizure/Epilepsy Type	Methods/Models	Main Findings	Potential Applications	Reference
Epilepsy (TLE patients)	Measurement of P2X7R radiotracer [3H]JNJ-64413739 in resected human brain tissue via PET.	No correlation between P2X7R radioligand uptake and age, sex, or the duration of epilepsy.	Diagnostic test to identify patients at riks of developing drug-refractory epilepsy after brain injury (e.g., hospitals). Identification of P2X7R expression in the brain.	[125]
Epileptogenesis (mice), Epilepsy (TLE patients)	PET imaging of P2X7R radiotracer ^18^F-JNJ-64413739 uptake in vivo in male mice subjected to intra-amygdala KA (0.3 µg) and ex vivo in resected tissue from TLE patients and control.	Increased P2X7R radiotracer uptake in the brain and peripheral organs according to the severity of SE in mice. Increased P2X7R radiotracer uptake in resected brain tissue from TLE patients.		[126]
Epileptogenesis (rats)	Longitudinal PET imaging of P2X7R radioligand ^18^F-FTTM in brain tissue of rats subjected to intrahippocampal KA (1.2 μL, 0.5 μg/μL).	Increased radiotracer uptake post-SE (e.g., hippocampus, amygdala, temporal cortex) which peaked during the latent period and which was mostly related to microglial activation.		[127]
Epilepsy (TLE patients and mice)	P2X7R protein levels measured via ELISA in plasma of patients with TLE and PNES. P2X7R protein expression in blood measured via FACS in EGFP-P2X7R reporter mice after intra-amygdala KA (0.3 µg) injections.	Increased P2X7R protein in plasma of patients with TLE when compared to control and patients with PNES. Increased P2X7R expression in blood cells (monocytes) post-SE.	Diagnostic tests to support stratification of patients at risk of epilepsy (e.g., general practitioner (GP) office, hospitals); identification of underlying inflammatory condition.	[128]
Febrile seizures (patients)	Association studies of genetic polymorphism in the *P2rx7* gene in infants with febrile seizures.	Association of gain-of-function missense rs208294 polymorphism in the *P2rx7* gene with susceptibility to childhood-onset febrile seizures.	Diagnostic test to identify infants at risk of febrile seizures.	[129]

Abbreviations: EGFP, enhanced green fluorescent protein; ELISA, enzyme-linked immunosorbent assay; FACS, Fluorescence-activated cell sorting; GP, general practitioner; KA, kainic acid; PNES, Psychogenic non-epileptic seizures;; TLE: Temporal lobe epilepsy.

## Data Availability

Not applicable.

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
