# Peer review of "The P2X7 Receptor as a Mechanistic Biomarker for Epilepsy"

_ijms, 2023, doi:10.3390/ijms24065410_

Round 1

Reviewer 1 Report

The manuscript titled "The P2X7 receptor as theranostic biomarker for epilepsy" submitted to IJMS (ijms-2249162), describes update findings regarding P2X7 receptor which has attracted particular attention as novel target for epilepsy treatment. Review is very well organized and written. Manuscript is divided properly and after comprehensive introduction including shortcomings in epilepsy treatment, author described very well cATP signalling via the P2X7R and the role of P2X7Rs during seizures and epilepsy.  Review is written very comprehensive way based on 140 references. An author cconcluded at the end that  that P2X7R-based treatments for epilepsy will reach the clinic within the foreseeable future, therefore topic is important and quite new.

Author Response

Thank you very much for the positive feedback. 

Reviewer 2 Report

This manuscript provides a very nice overview of the involvement of the P2X7 receptors in epilepsy and their potential use as therapeutic and diagnostic marker for this condition. Having said that, the current manuscript is somehow very similar to a recent article (Wong & Engel, 2023 Neuropharmacology; ref#23), which potentially may reduce its impact.

I have only minor concerns.

1) It would be useful to the readers if certain terminologies were defined. For instance, "theranostic", "mechanistic biomarker". 

2) The authors should not use the term "recent data" (l.96, l. 189) when referring to a citation that is a Review article (ref#23).   About 80 of the 140 citations are review articles without referencing original papers; it would be nice to include original citations whenever appropriate. The text on lines 122-124 is only attributed to ref#38, a review article. Text on lines 128-130 cites only Ref#40 (Dossi et al.,2021 Purinergic Signal), also a review article.

3) Pannexin-1 hemichannel should be Pannexin-1 channel.

4) Line 112. The extracellular concentration of ATP is at the nanomolar range, not micromolar as written.

5) The text (lines 129 - 134) is a little confusing with regard to citations, particularly refs# 40, 41 and 21. Please indicate which author used resected tissues from patients, which author measured extracellular ATP upon elevated K+, and which authors showed potent anticonvulsant effects of pannexin1 blockers.

6) Lines 155-162: Reasons why P2X7 receptors are attractive as therapeutic targets. The reason why some of the P2X7 characteristics are attractive (e.g., slow desensitization and membrane permeabilization) is missing from the text.

Author Response

Thank you very much for the positive feedback and for the suggested changes and improvements.

1) It would be useful to the readers if certain terminologies were defined. For instance, "theranostic", "mechanistic biomarker". 

The term mechanistic biomarker has now been defined in the text (Line 353):

“Moreover, mechanistic biomarkers, with a known role during disease pathogenesis, are much more informative and more accurately reflect the disease state compared to descriptive biomarkers which result as a side product of the disease.”

Because the term theranostic has not been mentioned within the review, this has been removed from the title and exchanged with mechanistic

“The P2X7 receptor as mechanistic biomarker for epilepsy”

2) The authors should not use the term "recent data" (l.96, l. 189) when referring to a citation that is a Review article (ref#23).   About 80 of the 140 citations are review articles without referencing original papers; it would be nice to include original citations whenever appropriate. The text on lines 122-124 is only attributed to ref#38, a review article. Text on lines 128-130 cites only Ref#40 (Dossi et al.,2021 Purinergic Signal), also a review article.

Recent data has been removed as suggested (line 110):

“Notably, data also suggest a diagnostic potential of the purinergic signalling system for seizures and epilepsy [23], making this signalling system particularly suited for the management of epilepsy and its treatment.”

According to the reviewers suggestion more original citations have been included were appropriate. This includes the sections regarding ATP release mechanisms (starting at line 128), P2X7 and Interleukin-1 (line 179) and ATP release during seizures and epilepsy (line 136).

As suggested, original citations have been included in addition to Ref number 40.

3) Pannexin-1 hemichannel should be Pannexin-1 channel.

As suggested, hemichannel has been changed to channel.

4) Line 112. The extracellular concentration of ATP is at the nanomolar range, not micromolar as written.

This has been corrected accordingly (Line 125):

“Usually present at low extracellular concentrations (micromolar nanomolar range)”

5) The text (lines 129 - 134) is a little confusing with regard to citations, particularly refs# 40, 41 and 21. Please indicate which author used resected tissues from patients, which author measured extracellular ATP upon elevated K+, and which authors showed potent anticonvulsant effects of pannexin1 blockers.

This has been changed accordingly:

“This includes evidence from a study where rat hippocampal slices were treated with the glutamate agonist ((S)-3,5-Dihydroxyphenylglycine) [45]. In another study, which was carried out in resected tissue from patients, Dossie et al. [21] showed that extracellular ATP concentrations increased by 80% during high K+-induced ictal discharges which was suppressed by blocking Pannexin-1. Of note, the same authors showed potent anticonvulsive effects when blocking Pannexin-1 during kainic acid (KA)-induced seizures in mice [21].”

6) Lines 155-162: Reasons why P2X7 receptors are attractive as therapeutic targets. The reason why some of the P2X7 characteristics are attractive (e.g., slow desensitization and membrane permeabilization) is missing from the text.

In the revised version, slow desensitization and membrane permeabilization has been removed:

“Among the P2XRs, the P2X7R has a relatively low affinity for ATP (EC50 ≥ 100 μM, activation threshold: 0.3-0.5 mM), suggesting P2X7R activation occurs mainly under conditions of high ATP release, restricted to the pathological focus. As a result, P2X7R-based treatments are hoped to lead to less adverse side effects when compared to current treatments (e.g., ASMs). Most importantly, however, the P2X7R is a key driver of inflammation [55-57]. P2X7Rs have been shown to contribute to the activation and proliferation of microglia [58] and are key regulators of the nucleotide-binding oligomerization domain-, LRR- and pyrin domain-containing protein 3 (NLRP3) inflammasome, inducing the release of inflammatory signalling molecules such as the pro-inflammatory cytokine interleukin-1β (IL-1β) [59-62].”